# Learning to Solve Nonlinear Partial Differential Equation Systems To Accelerate MOSFET Simulation

## Abstract

Semiconductor device simulation uses numerical analysis, where a set of coupled nonlinear partial differential equations is solved with the iterative Newton-Raphson method. Since an appropriate initial guess to start the Newton-Raphson method is not available, a solution of practical importance with desired boundary conditions cannot be trivially achieved. Instead, several solutions with intermediate boundary conditions should be calculated to address the nonlinearity and introducing intermediate boundary conditions significantly increases the computation time. In order to accelerate the semiconductor device simulation, we propose to use a neural network to learn an approximate solution for desired boundary conditions. With an initial solution sufficiently close to the final one by a trained neural network, computational cost to calculate several unnecessary solutions is significantly reduced. Specifically, a convolutional neural network for MOSFET (Metal-Oxide-Semiconductor Field-Effect Transistor), the most widely used semiconductor device, are trained in a supervised manner to compute the initial solution. Particularly, we propose to consider device grids with varying size and spacing and derive a compact expression of the solution based upon the electrostatic potential. We empirically show that the proposed method accelerates the simulation by more than 12 times. Results from the local linear regression and a fully-connected network are compared and extension to a complex two-dimensional domain is sketched.

## 1 Introduction

Nonlinear partial differential equations (PDEs) appear frequently in many science and engineering problems including transport equations for certain quantities like heat, mass, momentum, and energy (Fischetti & Vandenberghe, 2016). The Maxwell equations for the electromagnetic fields (Jackson, 1999), which govern one of the fundamental forces in the physical world, is one of the examples. By calculating the solution of those equations, the status of system-under-consideration can be characterized. In the machine learning society, solving a set of coupled partial differential equations has become an important emerging application field. (de Avila Belbute-Peres et al., 2020; Sanchez-Gonzalez et al., 2020)

Among many nonlinear partial differential equations, we consider the semiconductor device simulation (Grasser et al., 2003). The simulation is a pivotal application to foster next-generation semiconductor device technology at scale. Since the technology development heavily relies on the device simulation results, if the simulation time reduces, the turnaround time also significantly reduce. In order to reduce the simulation time, acceleration techniques based upon the multi-core computing have been successfully applied (Rupp et al., 2011; Sho & Odanaka, 2017). However, the number of cores cannot be exponentially increased and the cost also increases drastically as the number of cores involved increases. Moreover, as engineers submit many simulation jobs for a group of semiconductor devices, computing resources available to each simulation job is limited. As an alternative, we propose to improve the efficiency of the simulation *per se*.

In the semiconductor device simulation, a solution of a system of partial differential equations is numerically calculated with a certain boundary condition. Those differential equations are coupled

together and the overall system is highly nonlinear. The Newton-Raphson method (Stoer & Bulirsch, 2002) is known to be one of the most robust methods to solve a set of coupled nonlinear equations. When the method converges to the solution, the error decreases rapidly as the Newton iterations proceed. To achieve a rapid convergence, it is crucial that initial guess for the solution needs to be close enough to the real solution; otherwise, the method converges very slowly or may even diverge. Although we are interested in obtaining a solution at a specific boundary condition which is determined by an applied voltage, even obtaining an approximated solution to initiate the Newton-Raphson method successfully is challenging. In literature, in order to prepare an initial guess for the target boundary condition, several intermediate boundary conditions are introduced and solutions with those boundary conditions are computed sequentially (Synopsys, 2014). It, however, increases the overall computation time significantly. If the initial solution that is sufficiently close to the final one is provided by any means, we can save huge computational cost of calculating several unnecessary solutions.

Instead, we propose to learn an approximate initial solution of a set of coupled PDE for a target boundary condition by an artificial neural network. Specifically, when a set of labeled images is available, a neural network can be trained to generate a similar image for a given label. The trained model generates a numerical solution for a target boundary condition. We show that the proposed initial solution by our method can speed up the device simulation significantly by providing a better initial guess.

We summarize our contributions as follows:

- We derive a compact solution for PDE systems based on the electrostatic potential. As a result, the network size is reduced by a factor of three. Since the electrostatic potential is well bounded, the normalization issue can be avoided.
- For addressing various semiconductor devices, we propose a device template that can address various device structures with a set of hyper-parameters. Since the electrical characteristics of semiconductor devices are largely determined by the physical sizes of internal components, handling grids with varying size and spacing is particularly important.
- We propose a convolutional neural network (CNN) which generates the electrostatic potential from the device parameters. It can be used to accelerate the device simulation.
- Compared with the conventional method, the simulation time is significantly reduced (at least 12 times) with the proposed method while the numerical stability is not hampered.
- Results from the convolutional neural network are compared with other methods such as an alternative architecture (the fully-connected network) and the local linear regression.
- Our approach can be extended to the complex two-dimensional domains. Preliminary results are shown.

## 2 RELATED WORK

### 2.1 NEURAL NETWORKS FOR SOLVING DIFFERENTIAL EQUATIONS

Recently, there have been many attempts to build a neural network to solve a differential equation (Han et al., 2018; Long et al., 2018; Piscopo et al., 2019; Raissi et al., 2019; Winovich et al., 2019; Zhu et al., 2019; Obiols-Sales et al., 2020; Lu et al., 2020; Xiao et al., 2020). Among them, the Poisson equation is particularly of importance in the semiconductor device simulation. The Poisson equation plays a fundamental role in the semiconductor device simulation, by connecting the electrostatic potential and other physical quantities. In Magill et al. (2018), the Laplace equation (the Poisson equation with a vanishing source term) is considered for a nanofluidic device. A fully-connected neural network is trained to minimize the loss function, which combines the residue vector and the boundary condition. The authors assume a specific two-dimensional structure and the mixed boundary condition is applied. Another attempt to solve the Poisson equation is suggested in Özbay et al. (2019). The Poisson equation with the Dirichlet boundary condition is considered. It is decomposed into two equations. One is the Poisson equation with the homogeneous Neumann boundary condition. The other one is the Laplace equation with the Dirichlet boundary condition. A convolutional neural network architecture is adopted and the entire source term and the grid spacing are used as the input parameters. The network is trained with randomly generated source terms and

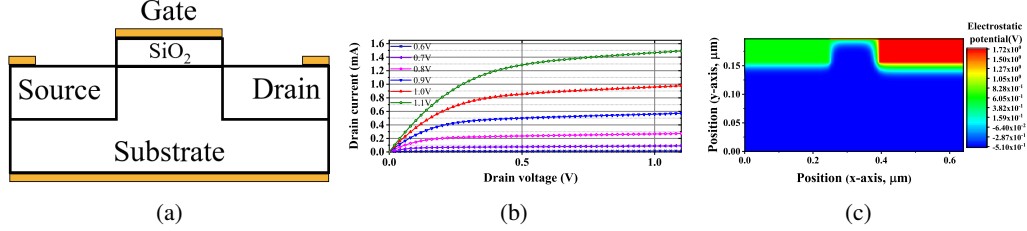

Figure 1: Components of the semiconductor device simulation. (a) Schematic diagram of a device with multiple terminals (yellow bars). (Input) (b) Current-voltage characteristics. (Output) (c) Internal quantity called the electrostatic potential. In order to calculate the terminal current, those internal quantities are needed.

a predicted solution shows a good agreement with the numerical solution. In order to consider a complex two-dimensional geometry, a rectangular domain covering the original geometry is introduced in Winovich et al. (2019) and a convolutional network is used. A neural network is used to solve the Poisson equation in a L-shape domain in Lu et al. (2020).

Quite recently, it has been independently proposed to accelerate the computational fluid dynamics simulations by predicting an appropriate initial guess from a convolutional neural network (Obiols-Sales et al., 2020). The basic idea is quite similar with this work and such a coincidence strongly demonstrates that the proposed approach is not restricted to a relatively narrow application domain.

## 2.2 NEURAL NETWORKS IN SEMICONDUCTOR DEVICE SIMULATION

Researchers are much interested with application of machine learning technique to the semiconductor device simulation (Carrillo-Nuñez et al., 2019; Bankapalli & Wong, 2019; Han & Hong, 2019; Souma & Ogawa, 2020). However, the actual research activities are quite diverse. In many cases, the neural networks are merely used as an efficient descriptor of the input-output relation without considering the internal physical quantities (Carrillo-Nuñez et al., 2019; Bankapalli & Wong, 2019). On the other hand, in their quantum transport simulation, Souma & Ogawa (2020) try to predict the electron density profile from the electrostatic potential. Although application of neural networks on the semiconductor device simulation is in its infancy, it has a huge potential.

## 3 PRELIMINARY: SEMICONDUCTOR DEVICE SIMULATION

We briefly introduce the semiconductor device simulation as a preliminary. As shown in Figure 1a, a multi-dimensional device structure has multiple terminals (yellow bars in the figure) whose voltages are controllable. Since the planar MOSFET has the translational symmetry along the direction perpendicular to the plane in Figure 1a, the three-dimensional device structure can be treated as a two-dimensional one. The terminal currents (in Ampere) under the determined terminal voltages (in Volt) are the output of the simulator, as shown in Figure 1b. As the motion of electrons yields the current, the terminal current itself can be only calculated from the internal physical quantities. (See Figure 1c.) Since the electrons are charged particles, the electric field due to the net charge density is also affected by the electronic motion. Therefore, we need to consider both effects of electronic motion and electric field by solving two equations, to calculate the terminal currents.

The first one is the continuity equation for electrons:

$$\nabla \cdot \mathbf{J}_n = 0, \tag{1}$$

where the electron current density vector, $\mathbf{J}_n$, is given by

$$\mathbf{J}_n = q\mu_n n\mathbf{E} + qD_n\nabla n, \tag{2}$$

where the elementary charge ($q$), the electron mobility ($\mu_n$), and the electron diffusion constant ($D_n$) are scalar parameters and the electron density ($n$) is a position-dependent unknown variable. Note that the current density ($\mathbf{J}_n$) also depends on the electric field vector ($\mathbf{E}$). Similar relations hold for holes with minor modification.

The second equation is the Gauss law, also called as the Poisson equation:

$$\nabla \cdot (\epsilon \mathbf{E}) = q(p - n + N_{dop}^+), \tag{3}$$

where $p$ is the hole density, $\epsilon$ is the permittivity (Jackson, 1999) and $N_{dop}^+$ is the positively charged impurity density. Under the electrostatic approximation, the electric field vector can be expressed in terms of the electrostatic potential ($\phi$) as

$$\mathbf{E} = -\nabla \phi. \tag{4}$$

With the electrostatic potential, the above set of equations has two unknown variables; $n$ and $\phi$. Note that nonlinearity is originated from (2). The first term in the right-hand-side of (2) is proportional to $n\mathbf{E}$, which is nonlinear with respect to unknown variables of $n$ and $\phi$. Therefore, when $\phi$ is fixed, the continuity equation becomes linear.

For numerical analysis, we need to discretize the equations. In general, the $i$-th component of the residual vector can be written as

$$r_i = f_i(\phi_1, ...\phi_N, n_1, ..., n_N, p_1, ..., p_N) - s_i = 0, \tag{5}$$

where $N$ is the number of grid points, $f_i(\cdot)$ is a nonlinear function, and $s_i$ is a constant. $s_i$ becomes nonzero only at the boundary nodes and depends on the voltages applied to the device terminals. Since the system is nonlinar, the solution at each boundary condition must be computed. Detailed discussion can be found in Appendix A.

The set of discretized equations in (5) is highly nonlinear and the iterative Newton-Raphson method is used to solve the system. Typically, a user wants to know the drain current at $V_G = V_D = V_{DD}$, where $V_{DD}$ is the maximum allowed voltage. However, an appropriate initial guess to start the Newton-Raphson method at that condition is not directly available. Only the solution at $V_G = V_D = 0$ V can be easily calculated. Starting from the solution with all zero terminal voltages, many intermediate steps toward the target terminal voltages are introduced. Based upon the converged solution at the previous condition as an initial guess, the next step is solved sequentially. Detailed discussion can be found in Appendix B. Therefore, computing several intermediate boundary values is the main source of increase of computation. Let us denote the number of boundary values simulated during the solution procedure as $N_{step}$. To evaluate the acceleration quantitatively, we define the reduction factor of the simulation time, $\alpha$, and it is well approximated as

$$\alpha \equiv \frac{\tau_{conv}}{\tau_{nn}} = \frac{N_{step} \times N_{newton}}{N_{newton}^{direct}} \approx N_{step}, \tag{6}$$

where $\tau_{conv}$ is the simulation time with the conventional method while $\tau_{nn}$ is the one with the proposed method. Moreover, $N_{newton}$ is the average number of the Newton iterations per a bias condition and $N_{newton}^{direct}$ is the number of the Newton iterations at the target bias condition. Since $N_{step}$ is typically larger than 10, we expect that the reduction factor can be larger than 10.

## 4 APPROACH

It is now clear that an approximate solution for a given boundary condition is a key to accelerate the device simulation. Instead of inventing yet another method to calculate the approximate solution efficiently, we propose a data driven approach by using a neural network to learn the numerical solution for desired boundary conditions.

### 4.1 COMPACT FORM OF A SOLUTION

In (5), the residual vector has $3N$ components, because we consider three internal quantities, $\phi$, $n$, and $p$, as solution variables. Typically, the number of grid points, $N$, ranges from thousands to tens of thousands. Therefore, it is desirable to introduce a compact form of a solution. Among various physical quantities in the semiconductor device, the electrostatic potential, $\phi$, is a key quantity. As discussed earlier, the nonlinearity arises from $n\mathbf{E} = -n\nabla\phi$ in (2). Under a fixed electrostatic potential profile, the electron and hole continuity equations become linear. The continuity equation decoupled from (3) is easy to solve. In other words, the electron and hole densities, $n$ and $p$, can be

readily obtained as long as a reasonably good potential profile is provided. Thus, we propose to use a neural network to generate electrostatic potential profiles.

By the compact form of the solution, the number of output components reduces by a factor of three. Moreover, the electrostatic potential is well bounded in a small voltage range and varies smoothly over the device structure; no normalization of the input data is required.

## 4.2 DEVICE TEMPLATE

The initial device design usually exhibits a sub-optimal performance as the device parameters are not optimal at the early stage of the technology development. During the technology development cycle, numerous structures are simulated for engineers to achieve better performance. Those devices are different in terms of the physical sizes of internal components and the doping densities. To enable our method to address various device types, we propose a device template of MOSFETs. By determining each hyper-parameter of the template, we simulate various types of MOSFETs. We illustrate a device template that we use in Figure 2.

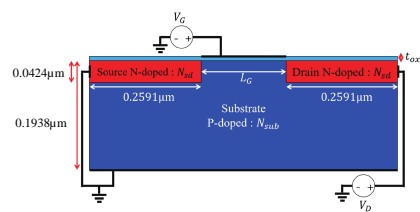

Figure 2: A proposed device template. By changing the hyper-parameters, one can readily simulate various types of MOSFETs.

Device parameters, such as the gate length ($L_G$), the oxide thickness ($t_{ox}$), and the doping concentrations ($N_{sd}$ for the source/drain regions and $N_{sub}$ for the substrate region) are varied within predefined ranges. For that purpose, the $x$-directional grid spacing under the gate terminal is adjusted and the proposed method can be applied to a set of various device structures whose grids have different spacing. Since the electrical characteristics of semiconductor devices are largely determined by the physical sizes of internal components, handling grids with varying size and spacing is particularly important.

## 4.3 NETWORK ARCHITECTURES

We propose a convolutional neural network to generate the two-dimensional potential profile for a given boundary condition. We design the architecture by adopting the generator part of the DCGAN (Radford et al., 2015) and illustrate it in Figure 3a. It takes the device parameters such as $L_G$, $t_{ox}$, $N_{sd}$, and $N_{sub}$ and the applied terminal voltages such as $V_G$ and $V_D$ as input. The output layer generates a 64-by-64 matrix as input for the simulation. In addition to the CNN, a fully-connected network (FCN) in Figure 3b is tested. Its depth is the same with the CNN and sizes of layers are designed to have a comparable level of computational complexity in the training phase.

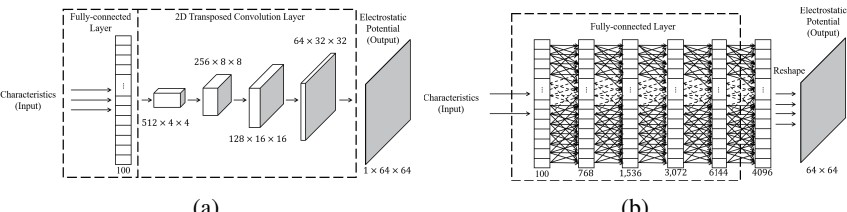

(a)                 (b)

Figure 3: Layer structures adopted in the two-dimensional MOSFET. (a) CNN and (b) FCN.

### 4.3.1 OBJECTIVE FUNCTION

To learn such network, we use and minimize a simple mean squared error objective function as

$$\mathcal{L} = \frac{1}{N} \sum_{i=1}^{N} \left( \tilde{\phi}_i - \phi_i \right)^2, \tag{7}$$

where $\tilde{\phi}_i$ is the electrostatic potential at the $i$-th node, predicted by the neural network. After 150 epochs, the mean square error of the electrostatic potential is sufficiently reduced.

## 5 EXPERIMENTS

### 5.1 EXPERIMENTAL SETUP

We conduct a set of experiments of MOSFET simulation (Taur & Ning, 1998). Device structures are shown in Figures 1a and 2. A two-dimensional grid with a size of 64-by-64 is used for all devices. Since the number of grid points is fixed, when the physical length of the MOSFET is changed, the grid spacing is also changed. For example, with a large $L_G$ value, the distance between two neighboring points near the gate terminal becomes large. Since $N$ is $64 \times 64 = 4,096$, the solution vector has $3N = 12,288$ unknown variables. We apply voltages to the gate terminal ($V_G$) and the drain terminal ($V_D$) to draw a current through the drain terminal ($I_D$). The room temperature is assumed to be 300 K throughout the experiments. We train our network in supervised fashion with the backpropagation algorithm on Pytorch 1.4.0 library. For the Adam optimizer, hyper-parameters such as the learning rate of $10^{-4}$, the L2 regularization coefficient of $10^{-7}$, and the batch size of 16, are used.

#### 5.1.1 DATASET

We use a simulator by Han & Hong (2019) as it is free from the license and the source code is publicly available. Detailed description on the simulator can be found in Appendix C. Each data point is specified with parameters of $L_G$, $t_{ox}$, $N_{sd}$, $N_{sub}$, $V_G$, and $V_D$ and their ranges are summarized in Appendix D. We curate the dataset with 10,112 instances that are randomly selected. We split the dataset by training, validation and test set by 70 % of the selected devices are the training set and the others in the validation (20 %) and test (10 %) sets. We will publicly release our splits for the future research in this avenue. The training and validation errors can be found in Appendix D.

#### 5.1.2 COMPARISON WITH THE CONVENTIONAL METHOD

As our primary goal is to accelerate the semiconductor device simulation with help of the initial solution obtained by the neural network, we evaluate the speed up of the device simulation algorithm compared to the conventional method. Specifically, we report the number of the Newton iterations for the converged solution at the target bias condition ($V_G = V_D = 1.1$ V). When the maximum potential update is smaller than $10^{-10}$ V, the convergence criterion holds.

By the conventional method, starting from the equilibrium condition of $V_G = V_D = 0.0$ V, we first increase the drain voltage up to 1.1 V. After that, the gate voltage is raised up to 1.1 V, which is called *bias ramping*. During the bias ramping, a uniform voltage step is applied and the solution at the previous boundary condition is used as the initial guess. When the voltage step is small enough, every simulation is successfully finished to get a final solution. However, with a large voltage step, some simulation runs may fail. In other words, the simulation is numerically unstable with a large voltage step.

We also report the ratio of the failed runs to the total simulation runs to check the numerical stability. In contrast, by our method, when the neural network provides the approximate solution, we start the simulation directly at the target bias condition. We compare the number of the Newton iterations for the converged solution with the one by the conventional bias ramping method with a uniform voltage step.

### 5.2 RESULTS FROM THE CNN

In this subsection, the results from the CNN are presented. After the training phase is finished, the trained neural network can be used to generate an approximate potential in the inference phase. Figure 4a shows the electrostatic potential profile generated by the trained neural network. The parameters of the MOSFET are $L_G = 96$ nm, $t_{ox} = 3.5$ nm, $N_{sd} = 4.8 \times 10^{20}$ cm$^{-3}$ and $N_{sub} = 1.8 \times 10^{18}$ cm$^{-3}$. The bias condition is $V_G = V_D = 0.24$ V. The potential difference between the predicted potential profile and the numerical solution, shown in Figure 4b, reveals that the maximum difference is around 0.1 V. The device in Figure 4a has a longer gate, $L_G = 164$ nm. Other parameters are also changed to $t_{ox} = 3.1$ nm, $N_{sd} = 2.1 \times 10^{20}$ cm$^{-3}$ and $N_{sub} = 2.0 \times 10^{18}$ cm$^{-3}$. Regardless $L_G$, the neural network can predict the electrostatic potential at a given bias condition ($V_G = 0.05$ V and $V_D = 0.66$ V) accurately.

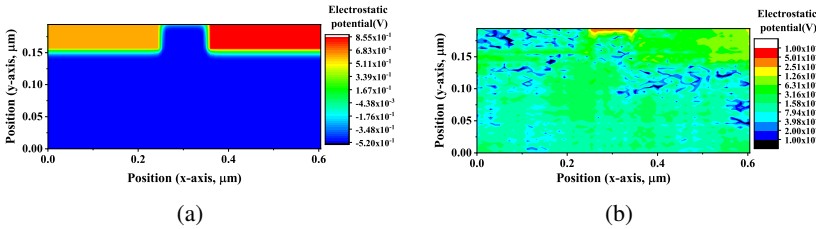

(a)                                             (b)

Figure 4: (a) Electrostatic potential profile predicted by the CNN at $V_G = V_D = 0.24$ V. (b) Difference between (a) and the numerical solution. The maximum potential error is about 0.1 V, which is reasonably small.

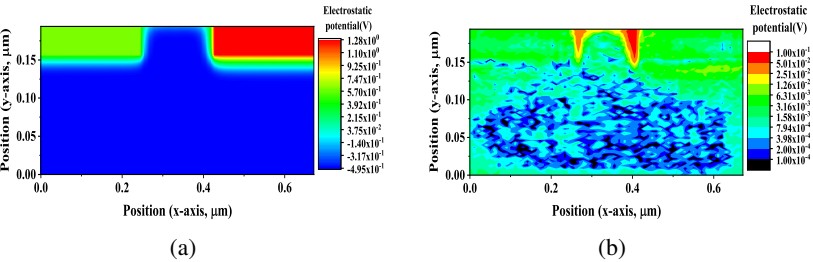

(a)                                             (b)

Figure 5: Same quantities with Figure 4 for a different MOSFET with $L_G = 164$ nm. $V_G = 0.05$ V and $V_D = 0.66$ V.

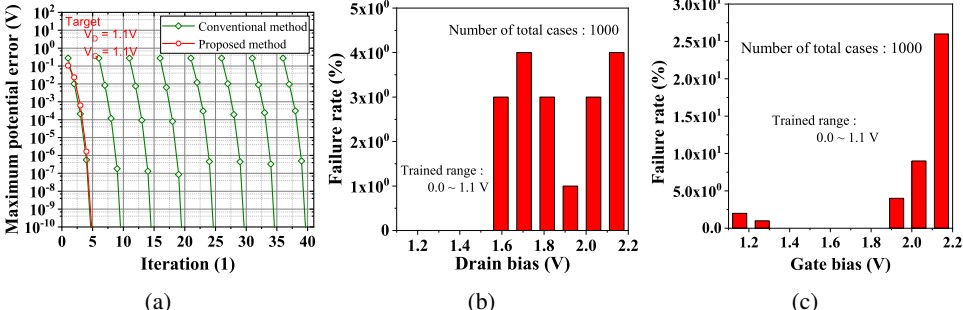

(a)                          (b)                          (c)

Figure 6: (a) Convergence behavior. (b) Failure rate for a drain voltage outside the training range. $V_D = 1.1$ V. (c) Failure rate for a gate voltage outside the training range. $V_G = 1.1$ V.

We demonstrate that the trained neural network generates the electrostatic potential profile close to the numerical solution in Figures 4 and 5. For 1,012 samples in the test set, average and variance of the maximum absolute potential error is 92 mV and 0.0016 $V^2$, respectively. The convergence behavior of the Newton-Raphson method is shown as a red curve in Figure 6a. With only five Newton iterations, the maximum potential update becomes smaller than $10^{-10}$ V. Note that as the initial update is quite small, about 0.1 V, the fast convergence is achieved. On the other hand, the conventional method with a uniform voltage step of 0.275 V (green curves) takes 40 iterations to reach the same result of our method. For each voltage condition, it takes only five iterations. However, after an intermediate boundary condition is solved, the next boundary condition should be solved again. Therefore, the green curves have peak values at every five iteration. Here, we achieve significant computation reduction factor of 8.

### 5.2.1 NUMERICAL STABILITY

In order to demonstrate the numerical stability of our method, the voltages beyond the training range are tested in Figures 6b and 6c. In Figure 6b, the drain voltage larger than 1.1 V is directly solved. For each drain voltage, 100 devices are generated and tested. Even for a high drain voltage of 2.2 V, the failure rate is just 4 %. We perform a similar test for the gate voltage larger than 1.1 V in Figure 6c. Although the failure rate increases sharply above 1.9 V, it remains quite small up to 1.8 V. These

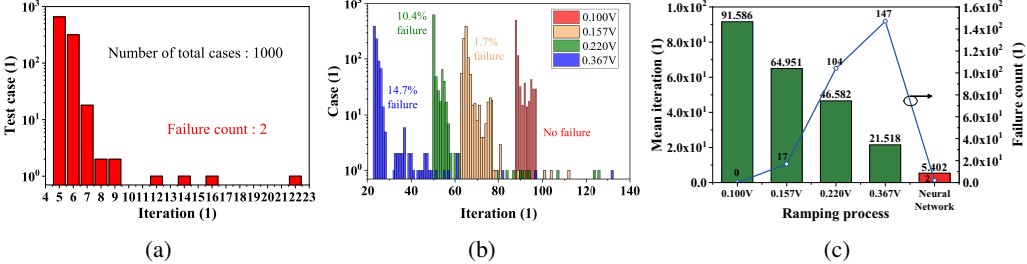

(a)                           (b)                           (c)

Figure 7: Numerical stability and speed of the proposed approach. (a) Distribution of the Newton iterations out of 1000 test cases. (b) Distribution of the Newton iterations with the conventional method. (c) Trade-off between the numerical stability and the simulation speed.

examples clearly demonstrate that the trained neural network can provide a sufficiently good initial potential profile.

### 5.2.2 Applications to Many Devices

We now investigate the applicability of our method to many devices. We evaluate our method on 1,000 different devices that are randomly generated. The simulation starts at $V_G = V_D = 1.1$ V. We show the distribution of the Newton iterations in Figure 7a. Among 1,000 test instances, only two samples fail to converge. For most cases (more than 99.7 %), four or five iterations are required for the converged solution. On the other hand, when the conventional method is applied, the number of the Newton iteration increases significantly as shown in Figure 7b.

In the conventional method, the voltage step is an important parameter to control the numerical stability and the speed. When we adopt a small voltage step of 0.1 V, there occurs no failure case. In this case, about 90 Newton iterations are needed to have the converged solution and the simulation is almost 20 times slower than the proposed approach. We may try to increase the voltage step for numerical efficiency. When the voltage step is 0.22 V, the number of the Newton iterations is reduced almost a factor of 2. However, the failure rate is larger than 10 %, which is certainly not tolerable. Even with a very large voltage step of 0.367 V, the number is 21, which is much larger than that of the proposed approach.

These observations are summarized in Figure 7c and Table 1. For the conventional method, there exists a trade-off between the numerical stability (related with the failure rate) and the simulation speed (related with the number of iterations). Our proposed approach exhibits both superior numerical stability and short simulation time. When the failure rate is limited up to 1%, the reduction factor of the simulation time is larger than 12.

The speedup is obtained with a cost of building the training dataset. If the number of inference runs is not large enough, the speedup by adopting the neural network will not yield any performance merit. However, it is noted that the semiconductor device simulator is typically developed by a developer group (a company or a research group) and distributed to several users. The simulation time experienced by the user is the most important metric to justify the additional cost of the developer group. Moreover, the total number of inference runs by the entire user group can easily exceed the number of data points in the training set. Similar cases may be found in other research fields, where the developer and user groups of the numerical simulators are well separated.

### 5.3 Comparison

As an alternative to the proposed neural network, we implement the local linear regression (Garimella) to predict an initial guess. For each prediction, we use the subset that has the closest dataset from input $\mathbf{x}_0$ with a ratio of $\beta$ to the whole dataset. The weight function is the tri-cube weight function,

$$w(d) = (1 - d^3)^3, \tag{8}$$

where $d$ is the normalized Euclidean distance. With cross-validation, we find that the fraction coefficient, $\beta$ is 0.2. We perform the same convergence test for many devices by using the local linear regression. As a result, the expected number of iterations is 6.5 and the failure rate is 2.1%. More

Table 1: Failure rate and speed of the conventional method and ours. $\alpha$ is the reduction factor defined in (6). The conventional method is tested with various voltage steps. For example, in a case of 0.1 V, 22 boundary conditions (11 for the drain sweep and 11 for the gate sweep) are solved. When the failure rate is too high, the second simulation round should start to solve the failed cases with a smaller voltage step, taking overall simulation time even longer. It implies that a high failure rate is not tolerable.

| | 0.1 V | 0.157 V | **CNN** (Ours) | FCN | 0.22 V | 0.367 V |
|---|---|---|---|---|---|---|
| Failure rate (%) | 0 | 1.7 | **0.2** | 2.6 | 10.4 | 14.7 |
| Number of iterations | 91.6 | 65.0 | **5.4** | 5.91 | 46.6 | 21.5 |
| $\alpha$ (Reduction factor (6)) | **17.0** | **12.0** | 1.0 | 1.09 | 8.6 | 4.0 |

results can be found in Appendix E. Clearly, the neural network outperforms the local linear regression in terms of the numerical stability. Moreover, in order to apply the regression, we need the densely-sampled dataset at each inference run. Since there exist several users, it is not practical to deliver the densely-sampled dataset to each user. The trained neural network is a more efficient way to approximate the densely-sampled dataset. In addition, another comparison is made between the CNN (Figure 3a) and the FCN (Figure 3b) in Table 1. Discussions can be found in Appendix F.

## 5.4 EXTENSION TO A COMPLEX TWO-DIMENSIONAL DOMAIN

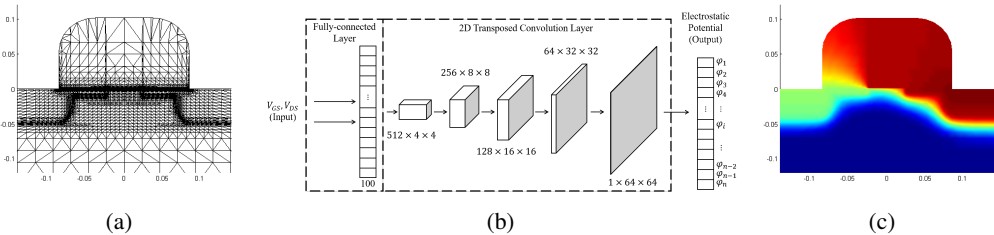

(a)                    (b)                    (c)

Figure 8: Complex two-dimensional domain. (a) An unstructured mesh for a MOSFET. (b) Layer structure of the adopted neural network. We have an internal CNN. (c) Predicted potential profile at $V_G = V_D = 1.0$ V. Colors represent the electrostatic potential.

Up to now, we have used a device template to describe various MOSFETs. In this subsection, a preliminary result about extension of our approach to a complex two-dimensional domain is presented. In Figure 8a, an unstructured mesh for a MOSFET is shown. A network shown in Figure 8b learns the electrostatic potential in supervised fashion. In this example, the output of the internal CNN in Figure 8b is mapped into the original unstructured mesh. The electrostatic potential is generated by the trained neural network, as shown in Figure 8c. Since the maximum potential error is lower than 76 mV, it can be used as an approximate solution. More information can be found in Appendix G.

## 6 CONCLUSION

We train a convolutional neural network to generate the electrostatic potential required for the semiconductor device simulation. By using the generated electrostatic potential as an initial guess, the target bias conditions can be simulated directly without a time-consuming ramping procedure. Our proposed approach can address a set of various device structures whose grids have different spacing. Moreover, we suggest a compact expression for the solution based upon the electrostatic potential to reduce the computational complexity.

In the empirical validations with two-dimensional MOSFETs, the simulation time has been significantly reduced (more than 12 times) with the proposed method while the numerical stability is not hampered. As deep neural networks perform well when a sufficient amount of data points are used in training, it is expected that the proposed approach would be accelerate the development cycle of semiconductors since massive simulation results are routinely generated during the technology optimization and are ready to be used for training such networks.

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

## A    DISCRETIZATION OF THE SEMICONDUCTOR EQUATIONS

Typically, the finite volume method (or the box method) (Laux & Grossman, 1985) is applied to convert the differential operator into the algebraic form. Then, the equation is converted into a collection of algebraic equations, which are evaluated at many sampling points (called vertex nodes). Since the original set is nonlinear with respect to $n$, $p$, and $\phi$, the converted algebraic equations are also nonlinear. The Newton-Raphson method is applied to the resultant set of equations. In this section, the discretization of the semiconductor equations, (1) and (3), is sketched.

It is observed that both equations have a common form of

$$\nabla \cdot \mathbf{F} - s = 0, \tag{9}$$

where $\mathbf{F}$ is the flux term and $s$ is the source term. In order to get a discretized form of the above equation at a certain grid point, we integrate the equation over a finite volume (or a box) surrounding the point. For example, in a grid shown in Figure 9, the discretized equation for the center point can be obtained by the integration over the red pentagon. With help of the divergence theorem (Jackson, 1999), we have

$$\oint_{Surface} \mathbf{F} \cdot d\mathbf{a} - \int_{Box} sd^3x = 0. \tag{10}$$

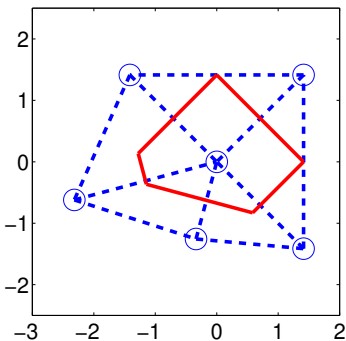

Figure 9: Grid points (Blue circles) and a finite volume surrounding the center point (A red pentagon).

Every node except for the terminal nodes contributes the residual vector following (10). We must also consider the boundary conditions at the terminal nodes. For the electron density ($n$) and the hole density ($p$), we have

$$n - n_{eq} = 0, \tag{11}$$

$$p - p_{eq} = 0, \tag{12}$$

where $n_{eq}$ and $p_{eq}$ are the carrier densities at the equilibrium condition. These densities do not depend on the applied terminal voltages. However, the boundary condition for the electrostatic potential ($\phi$) reads

$$\phi - \phi_{eq} - V_{terminal} = 0, \tag{13}$$

where $\phi_{eq}$ is the equilibrium potential and $V_{terminal}$ is the applied terminal voltage. When the applied terminal voltage changes, $s_i$'s in (5) at the terminal nodes change accordingly.

## B    BIAS RAMPING

The set of discretized equations in (5) is highly nonlinear and the iterative Newton-Raphson method is used to solve the system. Although a nonlinear relaxation scheme to avoid the full Newton-Raphson method (Meinerzhagen et al., 1991) has been proposed, the full Newton-Raphson method is still preferred due to its robustness.

Let us consider a typical case, where a user wants to know the drain current at $V_G = V_D = V_{DD}$. $V_{DD}$ is the maximum voltage which can be applied to the MOSFET. For example, $V_{DD}$ is 1.1 V in our example. The drain current at this bias condition is called as the ON current and the ON current represents the current-driving performance of the MOSFET. Since the ON current is of interest, it would be most desirable to calculate the ON current immediately. However, an appropriate initial guess to start the Newton-Raphson method at $V_G = V_D = V_{DD}$ is not available.

When $V_G = V_D = 0$ V, the device is in its equilibrium condition. In this special case, an effective way to generate the initial guess is well known (Jungemann & Meinerzhagen, 2003). It is based upon the charge neutrality condition, where $p - n + N_{dop}^+ = 0$ is imposed locally. Unfortunately, this method cannot be applied to non-zero terminal voltages. No established method for directly solving the equation at high terminal voltages exists. In order to overcome such a difficulty, the bias ramping technique is typically adopted. Starting from the equilibrium operating condition with all zero terminal voltages, many intermediate steps toward the target terminal voltages are introduced. Based upon the converged solution at the previous condition as an initial guess, the next step is solved with the Newton-Raphson method.

In this conventional solution method, the overall simulation time, $\tau_{conv}$, can be written as

$$\tau_{conv} = \sum_{i=1}^{N_{step}} \tau_{conv}^i = \sum_{i=1}^{N_{step}} N_{newton}^i \tau_{single} = N_{step} \times N_{newton} \times \tau_{single}, \tag{14}$$

where $N_{step}$ is the number of the entire bias conditions, $\tau_{conv}^i$ is the simulation time for the $i$-th bias condition, $N_{newton}^i$ is the number of the Newton iterations for the $i$-th bias condition, $\tau_{single}$ is the time spent for a single Newton iteration, and $N_{newton}$ is the average number of the Newton iterations. The bias ramping heavily increases the simulation time, because it introduces a large $N_{step}$.

In this work, we propose to solve the equation set at the target terminal voltages directly without the bias ramping. The overall simulation time with the proposed method based upon the neural network, $\tau_{nn}$, can be written as

$$\tau_{nn} = N_{newton}^{direct} \times \tau_{single}, \tag{15}$$

where $N_{newton}^{direct}$ is the number of the Newton iterations at the target bias condition. Two numbers, $N_{newton}$ and $N_{newton}^{direct}$, may be comparable. It is noted that $N_{step}$ in (14) does not appear any more in the above equation. The reduction factor of the simulation time, which is defined as $\tau_{conv}/\tau_{nn}$, can be approximated with $N_{step}$, as shown in (6).

## C  DEVICE SIMULATOR

In this work, the device simulator in Han & Hong (2019) has been used to obtain the simulation results. As it is free from the license issue, several simulation runs can be performed simultaneously. The Poisson equation, (3), the electron continuity equation, (1), and the hole continuity equation are coupled and solved self-consistently. A rectangular grid with varying spacing is adopted. Therefore, every finite volume for a grid point has four faces. The device template is implemented to accept the device parameters (such as $L_G$, $t_{ox}$, $N_{sd}$, and $N_{sub}$) as input parameters. When the Netwon-Raphson method is used, we must solve the following matrix equation:

$$\mathbf{J}\mathbf{u} = -\mathbf{r}, \tag{16}$$

where $\mathbf{J}$ is the Jacobian matrix, $\mathbf{u}$ is the update vector, and $\mathbf{r}$ is the residual vector. It is noted that the Jacobian matrix is very sparse in the semiconductor device simulation. In order to solve the above equation, a sparse matrix solver, the UMFPACK library (Davis, 2004), is used.

## D  MORE INFORMATION ON DATASET

More information on the dataset is provided.

Each data point is specified with parameters of $L_G$, $t_{ox}$, $N_{sd}$, $N_{sub}$, $V_G$, and $V_D$ and their ranges are summarized in Table 2.

Table 2: Ranges of input parameters

| Parameter | Range |
|---|---|
| Gate length, $L_G$ | 90 nm $\sim$ 170 nm |
| Oxide thickness, $t_{ox}$ | 2 nm $\sim$ 4 nm |
| Source/drain doping density, $N_{sd}$ | $5 \times 10^{19}$ cm$^{-3}$ $\sim$ $5 \times 10^{20}$ cm$^{-3}$ |
| Substrate doping density, $N_{sub}$ | $5 \times 10^{17}$ cm$^{-3}$ $\sim$ $5 \times 10^{18}$ cm$^{-3}$ |
| Gate voltage, $V_G$ | 0 V $\sim$ 1.1 V |
| Drain voltage, $V_D$ | 0 V $\sim$ 1.1 V |

We curate the dataset with 10,112 instances that are randomly selected. as shown in Figure 10a. The training and validation errors are measured as a function of the learning epoch as shown in Figure 10b.

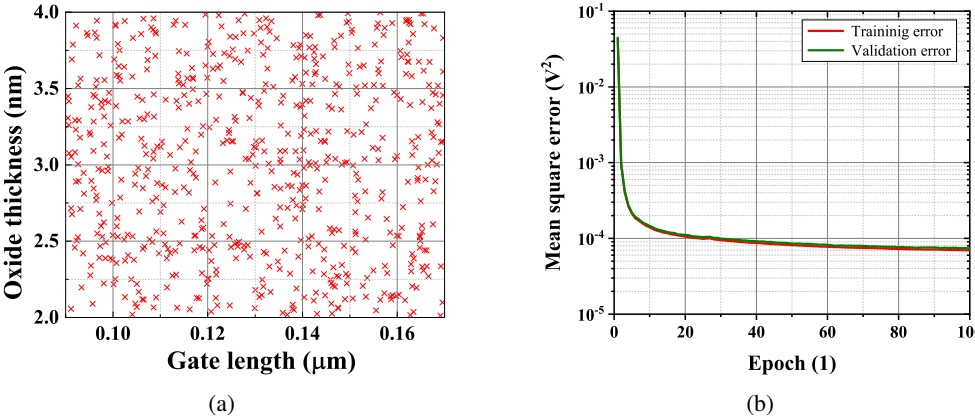

(a)  (b)

Figure 10: (a) Distribution of the selected devices in the $L_g$-$t_{ox}$ plane. (b) Training and validation errors as functions of the learning epoch.

# E    LOCAL LINEAR REGRESSION

We compare the distribution of the Newton iterations in Figure 11. Two methods (the CNN and the local linear regression) are compared. The failure rate of the local linear regression (2.1%) is much higher than that of the CNN (0.2%). When the failure rate is limited up to 1%, the regression does not meet the criterion. Moreover, the expected number of iterations (6.5) is also higher than that of the CNN (5.4).

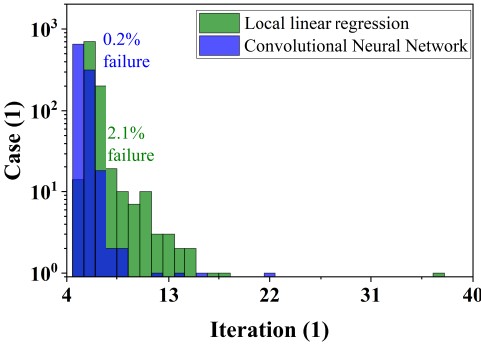

Figure 11: Distribution of the Newton iterations. Two methods (the CNN and the local linear regression) are compared.

## F  FULLY-CONNECTED GENERATIVE NETWORK

We compare the distribution of the Newton iterations in Figure 12. The expected number of iterations with the FCN (5.9) is larger than the one with the CNN (5.4). The failure rate of the FCN (2.6%) is much higher than that of the CNN (0.2%). Also the distribution in Figure 12 shows a long tail for the FCN, which indicates difficulties in the Newton-Raphson method. When the failure rate is limited up to 1 %, it is difficult to utilize the trained FCN, at least in its present form.

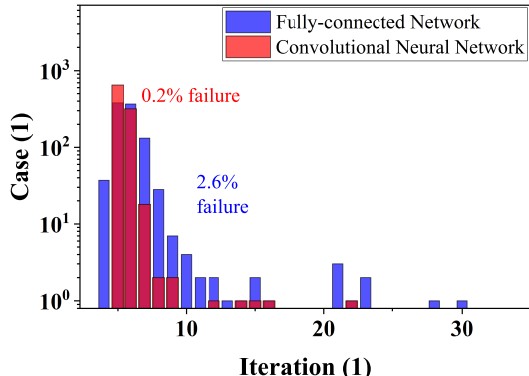

Figure 12: Distribution of the Newton iterations. Two methods (the CNN and the FCN) are compared.

It is noted that the FCN in Figure 3b is designed to have a comparable level of computational complexity in the training phase. When an approximate solution predicted by the FCN fails, the error at the very first Newton iteration tends to be larger than that of the CNN. Better performance enjoyed by the CNN can be attributed to the nature of the problem at hand. The neural network must generate the electrostatic potential profile in the two-dimensional domain. Since the potential profile is a smoothly varying function over the domain (as shown in Figure 4a and Figure 5a), it can be treated as an image, which is suitable for the CNN.

## G  MORE INFORMATION ON A MOSFET WITH AN UNSTRUCTURED MESH

In Figure 8a, a magnified view of the mesh structure near the channel region is shown. The entire mesh structure is shown in Figure 13. In total, 3,505 mesh points are assigned. The rounded regions at top describe the insulating layers, which separate different terminals.

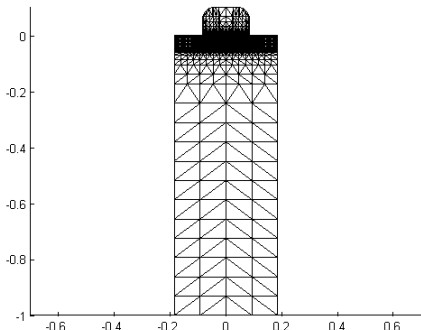

Figure 13: Entire mesh structure of the MOSFET in Figure 8a. Numbers in the axes represent coordinates in the micrometer unit.

The training and validation errors are measured as a function of the learning epoch as shown in Figure 14. After 100 epochs, the mean square error of the electrostatic potential is sufficiently reduced.

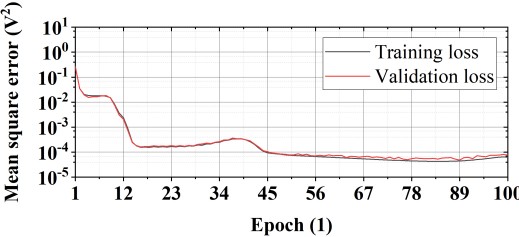

Figure 14: Training and validation errors as functions of the learning epoch.

We test the convergence behavior of the Newton-Raphson method at 121 bias points. For all of these bias points, the converged solutions are obtained.

With a uniform voltage step of 0.01 V, the total simulation time for a typical condition of $V_G = V_D$ = 1 V is 173 seconds. When the uniform step is increased to 0.05 V, it takes 60 seconds. However, with a uniform step of 0.1 V, it fails to converge. On the other hand, the simulation initiated by the trained neural network takes only 5 seconds. The speedup factor, $\alpha$, is larger than 10.

Finally, it is noted that the typical number of mesh points ranges from a few thousands (this example) to several dozens of thousands. Since the simulation time increases super-linearly with the mesh points, for a device with a dense mesh, the simulation time for a single run is typically expected to be several hours. Several simulation runs for optimization introduces a heavy computational burden.

