# OpenReview forum: "Learning to Solve Nonlinear Partial Differential Equation Systems To Accelerate MOSFET Simulation"
_ICLR.cc/2021/Conference — Reject_

### Official Review · AnonReviewer1 · 2020-10-27
**Use CNN to predict initial guess for the electrostatic potential in a Newton-Rhapson solution**

**Rating:** 4
**Confidence:** 4

**Review:**

The paper proposes to use a CNN to compute an initial guess for the iterative Newton-Rhapson solution of a coupled PDE system used for semiconductor device simulation. To do so, the authors construct a "device template" which parametrizes the design space. The CNN then maps a device configuration in this 6-dim space to the predicted electrostatic potential in the form of a 64x64 grid. The authors provide an analysis showing why predicting the electrostatic potential alone is sufficient. Overall, this approach can provide a simulation speedup of 12x or more. The authors are also planning to publish the dataset generated for the paper.

The idea to take a data-driven approach to compute the initial approximate solution in the context of the semiconductor simulation domain is interesting. Most prior work utilizing ML to accelerate PDE solution tried to generate such solutions directly. The path taken here, while seemingly easier, is interesting from an application point of view in that the solver is still used as before, potentially alleviating concerns about the validity of ML-accelerated solutions. The claimed order of magnitude speedup is significant (but see below for some comments on this). The main downside of the paper is the very limited novelty on the ML front (off-the-shelf generator from the DC-GAN paper) and demonstrated applicability to a specific and relatively narrow domain. As such, I recommend against publication at ICLR, and would advise the authors to submit the paper to a venue where the presented advances are likely to be of wider interest.

One specific point that I believe should be addressed, is to clarify the cost of building the dataset (10k simulations) and training the network, and its impact on the effective simulation speedup. As is, the 12x speedup would seem to apply in the limit of a very large number of network inference runs, and it is unclear if this is a practically relevant setting.

Suggestions for improvements:
- Explain to the reader explicitly why this is a 2d problem while the device exists in 3d space.
- What specifically was the validation set used for?
- In the text it is stated that "grid spacing is adjusted". Please explain more how this is done.
- The text states that the network was trained with backprop. Which specific optimizer was used, and with which hyperparameters?
- Including parts of the explanations currently in appendix B could make the text clearer and easier to read for people from outside of your field (e.g. the importance of bias ramping, and its relation to the simulation cost).

---

> ### Author Response · Authors · 2020-11-11
> **The simulation time experienced by the end user is important.**
>
> Thank you for your valuable comments. They are very useful in improving our manuscript. We are planning to improve the manuscript according to your comments. We will upload the revised manuscript by the end of the discussion stage 1 (November 17).
>
> In this time, point-by-point responses are provided.
>
> •	As pointed out by the reviewer correctly, it takes a long time to construct the training data set. Therefore, the hidden efforts to construct the training data set should be considered, if the device simulator is used only by the one who actually performs the training. We agree that it is not effective. However, it is noted that the semiconductor device simulator is typically developed by a developer group -- a company or a research group -- and distributed over several end users. The important metric is not the training time spent by the developer but the simulation time experienced by the end user.  For an end user, 12 times (or even larger) speedup is very attractive.
>
> •	Thank you for pointing out the 3D problem. The planar MOSFET has the translational symmetry along the third direction. In other words, every physical quantity does not change along the third direction, which is perpendicular to the plane. (See Figure 1(a) and 1(c), for example.) Of course, the state-of-the-art MOSFETs have complex 3D geometry. However, it is beyond our scope in this work. Coping the 3D MOSFET is a very challenging future research topic.
>
> •	According to many valuable suggestions by the reviewer for improvements, the revised manuscript will be prepared.
>
> Finally, thank you for your kind advice to submit our manuscript to other venues. Certainly, we will prepare a manuscript, which is more focused on the application aspect. However, we strongly believe that solving a set of coupled partial differential equations is an important emerging application field even from the machine learning society.

---

> ### Author Response · Authors · 2020-11-18
> **Many thanks to your detailed comments**
>
> The first revised manuscript is uploaded.
>
> In Subsubsection 5.2.2 of the revised manuscript, discussion on the cost of building the training set is made. The main observation is that the developer and user groups are well separated in this case and the simulation time is the most important metric. Certainly, in this application, a very large number of network inference runs is relevant.
>
> In Page 3, the reason why the three-dimensional structure can be treated as a two-dimensional one is explained. In Page 5, the meaning of “grid adjustment” is explicitly written. The optimizer and hyper-parameters are shown in Page 5. Some sentences are taken from Appendix B to make the text clearer.
>
> Thank you for your valuable comments.

---

> > ### Comment · AnonReviewer1 · 2020-11-25
> > **Thanks for addressing the comments**
> >
> > Thank you for updating the manuscript and addressing the comments from the review. I have raised my score by one point, but remain concerned about the expected narrow interest. While the acceleration of PDE solution is important and practically very useful, the paper demonstrates that this can be done in one very specific case. Given the vast possibilities of different PDE problems, I don't think the reader can confidently conclude that the results will generalize to some other domain.
> >
> > Regarding the cost of building the dataset, it would be ideal to be more quantitative about it, i.e. in terms of FLOPS and/or wall-time, how much does it cost to generate the training data and train the network in relation to a single inference run? This way, the reader could themselves make the judgement of whether the effort is worth it, based on the expected number of inferences they need for their problem of interest.

---

> > > ### Author Response · Authors · 2020-11-25
> > > **Thank you for your feedback!**
> > >
> > > Thank you for your feedback!
> > >
> > > Regarding your first comment, our approach can be applied to other fields, where a set of nonlinear equations must be solved in an iterative manner (the Newton-Raphson method).
> > > Although our examples are taken from a somewhat unfamiliar field (the semiconductor device simulation) from the viewpoint of the ML society, the basic idea is quite general and applicable to many unrelated fields.
> > > Moreover, we believe that it is an important task of the major ML conferences to find a new application field of the ML approach.
> > >
> > > At the end of Appendix G, the simulation time of each run is shown. The simulation time really depends on the mesh size. For a large device, typically hours are needed to get the solution. We hope that it can partially address your second comment.
> > >
> > > Finally, regarding the number of inference runs, our response to the Reviewer#2 is copied below. Rough guess for the total inference runs can be made:
> > >
> > > "The total number would be determined by a product between the number of users and the number of simulation runs per user. When we consider an academic research group, the user number would be 1 - 10 and the run number per user would be 10 - 100. Therefore, it is really a small number. However, when we consider a semiconductor company developing the next-generation technology, the number increases drastically. The user number would be (at least) 10 - 100. More importantly, an engineer can easily submit 100 - 1000 runs every day during the technology development period. Therefore, in the case of a semiconductor company, the additional cost spent by the developer group can be well justified."
> > >
> > > Thank you for your efforts in reviewing our manuscript.

---

### Official Review · AnonReviewer3 · 2020-10-28
**Accelerating semiconductor simulations with CNNs**

**Rating:** 5
**Confidence:** 2

**Review:**

The paper proposes a method for generating an initial guess for numerical simulations of semiconductor devices. Using CNNs to generate a better initial guess speeds up simulations by a significant constant factor.

Pros:
(1) The paper addresses an important practical problem with industrial applications.
(2) The exposition of the underlying theory and the presentation of results is very clear, the paper is well-written.
Even a non-expert like myself is able to follow the paper.
(3) The experimental results are promising.

Cons:
(1) Even though this is a good applications paper, the theoretical contributions to machine
learning are minimal -- and may not be enough to match ICLR standards.
(2) More justifications for the chosen architecture in Fig. 3, and better yet -- an experimental
comparison of alternative architectures would be interesting to see.

Overall, despite the promising experimental results, I am concerned that the paper may not meet
the standards of ICLR, hence my rating.

---

> ### Author Response · Authors · 2020-11-11
> **Alternative neural network architectures will be investigated.**
>
> Thank you for your valuable comments. Especially, your critical comment, “the theoretical contributions to machine learning are minimal,” seems to be very important. In order to meet the standards of ICLR, we are planning to improve the manuscript. In the revised manuscript, the following major improvements will be included:
>
> •	The linear regression result will be compared with our original result from the neural network.
>
> •	Alternative neural network architectures (other than the CNN) will be investigated.
>
> •	A new example for a complex 2D geometry will be prepared.
>
> We will upload the revised manuscript by the end of the discussion stage 1 (November 17).

---

> ### Author Response · Authors · 2020-11-18
> **An alternative architecture will be investigated in the second revised manuscript.**
>
> The first revised manuscript is uploaded.
>
> In Subsection 5.3 of the revised manuscript, results of the local linear regression are presented. Clearly, the neural network outperforms the local linear regression in terms of the numerical stability.
>
> In Subsection 5.4 of the revised manuscript, a new MOSFET with an unstructured mesh is considered.
>
> In order to include results of an alternative neural network architecture (other than the CNN), we are still working on it. We hope that the additional results can be found in the second revised manuscript.

---

> ### Author Response · Authors · 2020-11-24
> **A fully-connected network is investigated in the second revised manuscript.**
>
> The second revised manuscript is uploaded.
>
> In Figure 3(b), the fully-connected network tested in this work is shown. It is noted that the network has been designed to have a comparable level of computational complexity in the training phase. However, as shown in Table I and Appendix F, the CNN clearly outperforms the FCN. In Appendix F, we discuss the possible reason of such difference in the performance.
>
> Through the revision process in conjunction with the reviewers’ comments, we have addressed many issues. Major improvements include:
>
> •	Comparison with the local linear regression (Subsection 5.3 and Appendix E)
>
> •	Comparison with the fully-connected network (Subsection 5.3 and Appendix F)
>
> •	Results for a MOSFET with an unstructured mesh (Subsection 5.4 and Appendix G)
>
> •	Discussion on the efficiency of the proposed approach (Subsubsection 5.2.2)
>
> Thank you for your efforts in reviewing our manuscript.

---

### Official Review · AnonReviewer2 · 2020-10-31
**Unclear whether you need a convolutional neural network to solve a 6-input problem**

**Rating:** 6
**Confidence:** 4

**Review:**

The submission proposes speeding up MOSFET simulations by learning the electrostatic field by example.

On the positive side, the submission reports 12x speedup over running MOSFET simulation. The submission speeds up the simulation by reducing the problem to a lookup of a electrostatic field as a function of 6 parameters of a MOSFET (two doping levels, oxide thickness, the gate length, V_DD, and V_g).

However, the application doesn't seem to require a convolutional network.. the input space is only 6-dimensional, so simple techniques ought to work well. When I run into low-dimensional problems like this, I use local linear regression. [1] Namely:

1. Scale each input to lie in [0,1]
2. Choose a distance h in the 6-dimensional scaled input space (via cross-validating the following procedure)
3. For any test point, find N points that lie within distance h of the test point
4. Do weighted linear regression on those N points, with the tricube weighting function scaled to have support h [2]
5. Evaluate the linear regression once for each output position.

Note that in step 4, you form a NxN correlation matrix once, and perform a Cholesky decomposition on it. Then, for each position in the electrostatic output, you only need to do backsubstitution, which is O(N^2).

This would be the standard way of solving such a problem. You'd have to show that the neural network was substantially better or cheaper than this technique.

Another issue is with exactly how you measure the 12x speedup. That speedup time ignores the ~10,000 times you have to run the simulator to generate the training data. If the simulator is run only on the 1,000 test samples, then the simulator has actually slowed down by 10x (i.e., it's cheaper to not generate the training set and only simulate on the test samples). How many times is the simulator expected to run? If the simulator will run 100,000 times (e.g.), then it could be that the 5-d space is sampled densely enough where you can simply perform a nearest neighbor computation and return the electrostatic field out of memory, and do absolutely no interpolation or simulaion.

References.
1. https://en.wikipedia.org/wiki/Local_regression
2. https://en.wikipedia.org/wiki/Kernel_(statistics)

---

> ### Author Response · Authors · 2020-11-11
> **The local linear regression will be tried.**
>
> Thank you for your valuable comments. Your two comments are very useful in improving our manuscript. We are planning to improve the manuscript according to your comments. We will upload the revised manuscript by the end of the discussion stage 1 (November 17).
>
> In this time, point-by-point responses are provided.
>
> •	Following the first comment, the local linear regression over the 6-dimensional input space (two doping densities, two lengths, and two bias voltages) will be tried and its results will be compared with our original results in the revised manuscript. Thank you for your suggestion.
>
> •	For the second comment, as pointed out by the reviewer correctly, it takes a long time to construct the training data set. Therefore, the hidden efforts to construct the training data set should be considered, if the device simulator is used only by the one who actually performs the training. We agree that it is not effective. However, it is noted that the semiconductor device simulator is typically developed by a developer group -- a company or a research group -- and distributed over several end users. Although the number of simulation runs by a single end user can vary depending on his/her need, the total number would be very large. Moreover, since we are interested with the end users, the dense sampling of the input space is not feasible. Therefore, it is desirable for the developer group to distribute a semiconductor device simulation with a trained neural network. The above discussion will be added in the revised manuscript, in order to clarify the motivation of this work.

---

> > ### Comment · AnonReviewer2 · 2020-11-24
> > **Thanks for trying local linear regression**
> >
> > Thanks to the authors for trying local linear regression: it's good to see that the convolutional neural network beats that benchmark (although it would be good to understand why). It's also good to see the preliminary results in Appendix F on non-templated MOSFETs (although there aren't enough details to understand what's going on).
> >
> > I think the qualitative comments about usage partially address my question on running the simulator 10,000 times. It would be better to have some sort of estimate of the number of times such a simulator would be run.
> >
> > I suspect that this paper will have relatively narrow appeal in ICLR. I'll raise my score from a 4 to a 6.

---

> > > ### Author Response · Authors · 2020-11-24
> > > **Thank you for your feedback!**
> > >
> > > Thank you for your response about the first revised manuscript. We have just uploaded the second revised one. In the present manuscript, the results of a fully-connected network are added in Subsection 5.3 and Appendix F. We hope that addition of these results further improves the quality of our manuscript.
> > >
> > > Point-by-point responses are provided:
> > >
> > > •	(although it would be good to understand why) Thank you for your comment. We have the following plan. We will pick up several cases, where the CNN is successful but the local linear regression is not. Then, by comparing them, the underlying reason will be investigated.
> > >
> > > •	(It would be better to have some sort of estimate of the number of times such a simulator would be run.) The total number would be determined by a product between the number of users and the number of simulation runs per user. When we consider an academic research group, the user number would be 1 - 10 and the run number per user would be 10 - 100. Therefore, it is really a small number. However, when we consider a semiconductor company developing the next-generation technology, the number increases drastically. The user number would be (at least) 10 - 100. More importantly, an engineer can easily submit 100 - 1000 runs every day during the technology development period. Therefore, in the case of a semiconductor company, the additional cost spent by the developer group can be well justified.
> > >
> > > Thank you for your efforts in reviewing our manuscript.

---

> ### Author Response · Authors · 2020-11-18
> **Subsection 5.3 is added and Subsubsection 5.2.2 is modified.**
>
> The first revised manuscript is uploaded.
>
> In Subsection 5.3 of the revised manuscript, following your valuable comment, results of the local linear regression are presented. The local linear regression really works (as expected) and its results are better than the conventional method. However, the neural network outperforms the local linear regression in terms of the numerical stability. More importantly, the local linear regression requires the densely-sampled dataset at each inference run. We believe that the trained neural network is a more efficient way.
>
> In Subsubsection 5.2.2 of the revised manuscript, discussion on the cost of building the training set is made. The main observation is that the developer and user groups are well separated in this case and the simulation time is the most important metric.
>
> Thank you for your valuable comments.

---

### Official Review · AnonReviewer5 · 2020-11-04
**Mehtod is not new; good engineering application**

**Rating:** 5
**Confidence:** 5

**Review:**

The paper proposed to use a CNN as a low-accuracy solver to output a good initial guess from the device parameters and then the solution is refined by an expensive numerical solver to get a good accuracy. The experiment on the MOSFET problem shows that, compared with the conventional numerical solver, this hybrid strategy can accelerate the simulation by more than 12 times.

Pros:
- The proposed hybrid method speeds up the simulation a lot without a sacrifice of accuracy.

Major comments:
- This idea of using CNN to output a good initial guess is not new and has been proposed, e.g., in the paper https://arxiv.org/abs/2005.04485 . So there is little contribution in the method.
- The problem considered in the paper is a 2D rectangle domain, which can be handled by CNN nicely. However, in practice this may not be true. The authors should consider more complex domains. For example, in https://doi.org/10.1016/j.jcp.2019.05.026 CNN is used for complex 2D geometry. In https://arxiv.org/abs/1907.04502 neural network is used to solve Poisson equation in a L-shape domain, which should also be discussed in Section 2.1.
- This paper has no contribution in machine leaning, and instead it is an application for solving PDEs. Most part of this paper is not relevant to machine learning topics. It may be more suitable to publish this paper in an engineering journal instead of a machine learning conference.

---

> ### Author Response · Authors · 2020-11-11
> **Results for a complex 2D geometry will be prepared.**
>
> Thank you for your valuable comments. They are very useful in improving our manuscript. We are planning to improve the manuscript according to your comments. We will upload the revised manuscript by the end of the discussion stage 1 (November 17).
>
> In this time, point-by-point responses are provided.
>
> •	Thank you for introducing a recent publication, https://arxiv.org/abs/2005.04485. Although we did not know it, it will be very helpful. In the revised manuscript, it will be added in Section 2. And it is noted that the computational fluid dynamics and the semiconductor device simulation are much different research fields. Therefore, we believe that this work still has its merits.
>
> •	In the revised manuscript, https://doi.org/10.1016/j.jcp.2019.05.026  and https://arxiv.org/abs/1907.04502 will be discussed in Section 2.
>
> •	Thank you for pointing out the problem related with complex domains. As correctly pointed out by the reviewer, application to the complex 2D domain is an important issue. We admit that our CNN approach is not very suitable to deal with the complex 2D domain. However, we believe that our approach is not restricted to a specific neural network architecture. In order to demonstrate this point, another MOSFET with a complex 2D geometry will be additionally studied in the revised manuscript. We hope that the new example convinces the reviewer that our approach is applicable to more general cases.
>
> Finally, thank you for your kind suggestion to submit our manuscript to an engineering journal. Certainly, we will prepare a manuscript, which is more focused on the application aspect. However, we strongly believe that solving a set of coupled partial differential equations is an important emerging application field even from the machine learning society.

---

> > ### Comment · AnonReviewer5 · 2020-11-25
> > **Thanks for the revision.**
> >
> > Thanks for revising the paper by adding a new case of non-rectangle geometry in Section 5.4. However, my main concern is still that the ML part of this paper is very standard, and has no contribution in ML.

---

> > > ### Author Response · Authors · 2020-11-25
> > > **Thank you for your feedback!**
> > >
> > > Thank you for your feedback!
> > >
> > > In the second revised manuscript, we have Subsection 5.3, where comparison with other methods is made.
> > > The local linear regression and the fully-connected network are considered and the CNN clearly outperforms them in terms of the numerical stability.
> > > Two appendices (Appendix E and Appendix F) are provided to show detailed information about them.
> > >
> > > Of course, this work does not introduce a new network architecture to solve the problem at hand.
> > > However, the benchmark against baselines has been newly added during the revision process, as shown in the above paragraph.
> > >
> > > Instead, it demonstrates a new application field (the semiconductor device simulation) and a clear performance improvement (at least one order-of-magnitude).
> > > We believe that it is an important task of the major ML conferences to find a new application field of the ML approach.
> > > Nowadays, we can easily find many application papers, whose application field is the computational fluid dynamics.
> > > Quite similarly, another application field which enjoys the advantages of the ML approach can be considered in the ML conferences.
> > >
> > > Finally, we believe that our approach can be applied to other fields, where a set of nonlinear equations must be solved in an iterative manner.
> > >
> > > Thank you for your efforts in reviewing our manuscript.

---

> ### Author Response · Authors · 2020-11-18
> **Subsection 2.1 is modified and Subsection 5.4 is added.**
>
> The first revised manuscript is uploaded.
>
> In Subsection 2.1 of the revised manuscript, the references mentioned in the reviewer comment are introduced. And a new paragraph for https://arxiv.org/abs/2005.04485 is added.
>
> In Subsection 5.4 of the revised manuscript, a new MOSFET with an unstructured mesh is considered.
>
> Thank you for your valuable comments.

---

### Official Review · AnonReviewer4 · 2020-11-05
**Authors propose using a ANN to learn an approximate solution for desired boundary conditions in order to accelerate the semiconductor device simulation, They show that the computational cost is significantly reduced. They empirically show that their method accelerates simulation by more than 12 times.**

**Rating:** 7
**Confidence:** 4

**Review:**

Contribution: Authors propose using a neural network to learn an approximate solution for desired boundary conditions in order to accelerate the semiconductor device simulation. They significately reduce the computational cost to calculate several unnecessary solutions when considering an initial solution sufficiently close to the final one by a convolutional neural network (CNN). To compute this initial solution, authors authors train a MOSFET based convolutional neural network. They empirically show that their proposed method accelerates the simulation by more than 12 times.

Pros:

1) Proposal is original.
2) Proposal is fast.

---

> ### Author Response · Authors · 2020-11-11
> **We will prepare a revised manuscript.**
>
> Thank you for your favorable comments. In order to address many valuable comments by other reviewers, we are planning to improve the manuscript. In the revised manuscript, the following major improvements will be included:
>
> •	The linear regression result will be compared with our original result from the neural network.
>
> •	Alternative neural network architectures (other than the CNN) will be investigated.
>
> •	A new example for a complex 2D geometry will be prepared.
>
> We will upload the revised manuscript by the end of the discussion stage 1 (November 17).

---

### Author Response · Authors · 2020-11-18
**The first revised manuscript is uploaded.**

Dear Reviewers and Area Chair of Paper749,

The first revised manuscript is uploaded. In this revised manuscript, we try to resolve several issues raised by the reviewers. Modified sentences are written in the blue letters. In the revised manuscript, the appendices are attached to the main document.

The major improvements are as follow:

•	Results of the local linear regression are obtained. (Subsection 5.3) Clearly, the neural network outperforms the local linear regression in terms of the numerical stability.

•	A new MOSFET with an unstructured mesh is considered. (Subsection 5.4)

•	Discussion on the cost of building the training set is made. (Subsubsection 5.2.2)

•	Two additional appendices (Appendices E and F)

We are currently preparing the second revised manuscript, in order to include results of an alternative neural network architecture (other than the CNN).

Best regards,

Authors of Paper749

---

### Author Response · Authors · 2020-11-24
**The second revised manuscript is uploaded.**

Dear Reviewers and Area Chair of Paper749,

The second revised manuscript is uploaded. Modified sentences are written in the blue letters. In the second revised manuscript, the appendices are attached to the main document.

The major improvements in the second revised manuscript are as follow:

•	Results of the fully-connected network are added. (Figure 3(b), Table I, and Appendix F)

•	One appendix for the fully-connected network is added (Appendix F) and one existing appendix for the unstructured mesh is updated (Appendix G)

The present authors believe that our manuscript has been improved significantly through the revision process in conjunction with the reviewers’ comments. All issues raised by the reviewers (including the local linear regression, an alternative neural network architecture, an unstructured mesh, and efficiency of the proposed approach) are addressed.  Thank you for your valuable comments.

Best regards,

Authors of Paper749

---

### Decision · Program_Chairs · 2021-01-07
**Final Decision**

**Decision:**

Reject

**Comment:**

This paper proposes using a neural network to learn an approximate solution for desired boundary conditions to accelerate the semiconductor device simulation. The work shows that speed-up simulation is increased significantly. However, the major concern about this work is the limited contribution to the machine learning community as exposed by the reviewers.